# Zero-CL: Instance and Feature decorrelation for negative-free symmetric contrastive learning

**Shaofeng Zhang[1], Feng Zhu[2], Junchi Yan[1]\*Rui Zhao[1,2], Xiaokang Yang[1]**
[1]Shanghai Jiao Tong University, [2]SenseTime Research
`{sherrylone, yanjunchi, xkyang}@sjtu.edu.cn`
`{zhufeng, zhaorui}@sensetime.com`

## Abstract

For self-supervised contrastive learning, models can easily collapse and generate trivial constant solutions. The issue has been mitigated by recent improvement on objective design, which however often requires square complexity either for the size of instances ($\mathcal{O}(N^2)$) or feature dimensions ($\mathcal{O}(d)^2$). To prevent such collapse, we develop two novel methods by decorrelating on different dimensions on the instance embedding stacking matrix, i.e., **I**nstance-wise (ICL) and **F**eature-wise (FCL) **C**ontrastive **L**earning. The proposed two methods (FCL, ICL) can be combined synthetically, called Zero-CL, where "Zero" means negative samples are **zero** relevant, which allows Zero-CL to completely discard negative pairs i.e., with **zero** negative samples. Compared with previous methods, Zero-CL mainly enjoys three advantages: 1) Negative free in symmetric architecture. 2) By whitening transformation, the correlation of the different features is equal to zero, alleviating information redundancy. 3) Zero-CL remains original information to a great extent after transformation, which improves the accuracy against other whitening transformation techniques. Extensive experimental results on CIFAR-10/100 and ImageNet show that Zero-CL outperforms or is on par with state-of-the-art symmetric contrastive learning methods.

## 1 Introduction

One of the current main bottlenecks in deep network training is the dependence on heavy annotated training data, and this motivates the recent surge of interests in unsupervised (Donahue & Simonyan, 2019) and self-supervised (Chen & He, 2021; Chen et al., 2020) methods. Specifically, in self-supervised representation learning (SSL), a network is pre-trained without any form of manual annotation, thus providing a means to extract information from unlabeled data sources (e.g., text corpora, videos, images from the Internet, etc.). In self-supervision, label-based information is replaced by a prediction problem using a certain context or using a pretext task. Pretext task in SSL can mainly be divided into three categories: **1) Generative based** approaches (Donahue & Simonyan, 2019) learn to generate or otherwise model pixels in the input space. However, pixel-level generation is computationally expensive and may not be necessary for representation learning. **2) Contextual based** methods (Vincent et al., 2008; Pathak et al., 2016; Ye et al., 2019) design pretext tasks (denoising auto-encoders (Vincent et al., 2008), context auto encoders (Zhang et al., 2016; 2017), etc). **3) Contrastive based** methods (Chen et al., 2020; Grill et al., 2020; Caron et al., 2020; Asano et al., 2019) take augmented views of the same image as positive pair and others as negative pairs. Generally, one positive sample corresponds to lots of negative samples. In recent works, contrastive based methods have shown great promise, achieving state-of-the-art results in image classification (Chen et al., 2020), video classification (Han et al., 2020) and other downstream tasks (Chen & He, 2021).

However, trivial constant solutions (different samples get the same representation) is easily happening without the proper design of architecture and objective function. The well-known solutions to

---

\*Junchi Yan is the correspondence author. Shaofeng Zhang, Junchi Yan, Xiaokang Yang are also with MoE Key Lab of Artificial Intelligence, Artificial Intelligence Institute, Shanghai Jiao Tong University. Rui Zhao is also with Qing Yuan Research Institute, Shanghai Jiao Tong University.

avoid this problem can be summarized into two parts: asymmetric model architecture and proper objective function. **1) Model architecture:** MoCo (He et al., 2020), BYOL (Grill et al., 2020) update encoders separately and stopping gradient operation is adopted to avoid such problem. Then, BYOL and SimSiam (Chen & He, 2021) introduce a predictor module to avoid collapse, which is composed by MLP (Goodfellow et al., 2016). Current mainstream interpretation is using the predictor to construct an asymmetric structure, which is useful to alleviate trivial solutions. **2) Objective function:** SimCLR uses symmetric framework in contrastive learning. They prevent trivial solutions by using negative pairs and InfoNCE, where InfoNCE can be divided into an alignment term and a uniformity term (Arora et al., 2019). The uniformity term pulls different samples to a hyper-sphere uniformly, forcing obtaining different representations and avoiding trivial solutions. Recently, Barlow Twins (Zbontar et al., 2021) designs a new objective function from the information redundancy perspective, which also has two terms (an invariance term and a redundancy reduction term). The invariance term maximizes the correlation of the same feature across different views, and the redundancy term reduces information redundancy. However, for the symmetric framework, both SimCLR and Barlow Twins require square order complexity in objective functions, and the main complexity comes from the uniformity and redundancy term. In this paper, we propose two methods, named Zero-ICL and Zero-FCL, where Zero-ICL discards the uniformity term and only requires $\mathcal{O}(N)$ complexity by instance-wise whitening. Correspondingly, Zero-FCL discards the redundancy term by feature-wise whitening and only requires $\mathcal{O}(d)$ complexity. **Our contributions are**:

**1)** We propose a new contrastive learning framework to prevent trivial solutions, Zero-CL, which includes two parts, i.e., Zero-ICL (instance-wise) and Zero-FCL (feature-wise), either of which can work independently and only requires linear order complexity (objective function).

**2)** To our best knowledge, Zero-ICL is the first attempt of instance-wise whitening, which is conceptually comprehensible for preventing collapses in contrastive learning. Note that most previous methods (including other domains beyond vision) e.g. (Eldar & Oppenheim, 2003; Kessy et al., 2018) only use whitening transformation to reduce the information redundancy on feature-wise.

**3)** We give empirical analysis on the relationship between previous methods and our Zero-CL, where previous negative sample consuming methods (Zbontar et al., 2021; Chen et al., 2020) can be regarded as our method with Lagrangian transformation. Then, we theoretically introduce ZCA-based whitening from the *maximal correlation* (Kessy et al., 2018) perspective.

**4)** Experimental results on standard image benchmarks (CIFAR-10/100 and ImageNet-100/1k) show our method achieves new state-of-the-art results for symmetric contrastive learning compared with (Chen et al., 2020; Zbontar et al., 2021), especially for small hidden dimension and batch size.

## 2 RELATED WORK

We review recent contrastive learning works. We mainly divide the previous contrastive methods into **feature-wise** and **instance-wise** according to different contrastive dimensions. We particularly discuss the similarity and differences of some most related methods in Section 5 in detail.

**instance-wise contrastive learning.** instance-wise contrastive learning aims to attract positive pairs and repulse negative pairs (Chen et al., 2020; He et al., 2020), where each pair is composed of two instance views. One of the widely used objective functions in instance-wise learning is InfoNCE, and extensive experiments in (Oord et al., 2018) show that InfoNCE requires negative pairs to avoid trivial solutions. SimCLR (Chen et al., 2020) regards views augmented from different images in a mini-batch as negative pairs. However, SimCLR requires a large batch size to improve the accuracy, which is GPU intensive. To address this issue, MoCo (He et al., 2020) proposes storing negative samples in the memory bank and updates the bank by the first-in-first-out principle. MoCo further proposes two operations to prevent trivial solutions, i.e., momentum update key encoder and stop gradient. In recent works (Chen et al., 2020; Grill et al., 2020), the mainstream explanation of the two operations is they construct an asymmetric framework, which can avoid trivial solutions. Inspired by this, BYOL (Grill et al., 2020) adopts stop gradient operation and EMA algorithm to update target encoder. Further, BYOL introduces a predictor module to build a more asymmetric framework. Then, SimSiam (Chen & He, 2021) empirically shows that stop gradient and the predictor is the key component to avoid trivial solutions. Another approach to preventing such solutions is introduced in W-MSE (Ermolov et al., 2021). Different from previous works (Chen et al., 2020; Chen & He, 2021; He et al., 2020), W-MSE requires multiple augmented views to improve the accuracy. Besides, W-

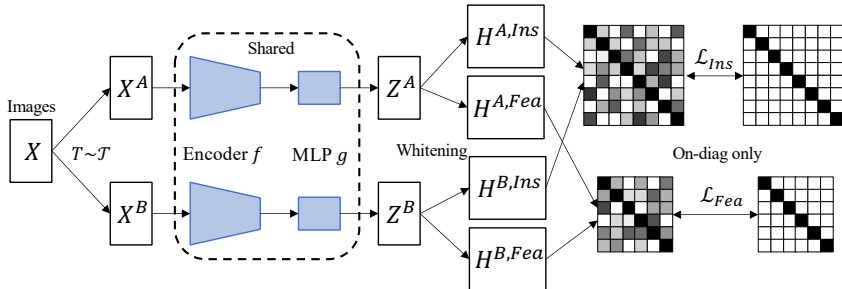

Figure 1: Framework of the proposed Zero-CL. The illustrated embedding $\mathbf{Z} \in \mathbb{R}^{8 \times 6}$ means there are eight samples with six features. Zero-CL includes two branches, i.e., feature-wise branch (Zero-FCL, bottom) and instance-wise branch (Zero-ICL, top). In instance-wise branch, whitening transformation is conducted on instance dimension. Then, the trace of cross affinity matrix $\mathrm{tr}\big((\mathbf{H}^A)^\top \mathbf{H}^B\big)$ is maximized. Note feature-wise branch is similar to instance-wise branch.

MSE simply borrows Cholesky-based whitening transformation in SSL and the multi-view objective requires $\mathcal{O}(NM^2)$ complexity, where $M$ is the number of views.

**feature-wise contrastive learning.** Different from instance-wise learning, feature-wise methods contrast on feature dimension. Barlow Twins (Zbontar et al., 2021) is the first feature-wise method, which pushes the cross-correlation matrix to identity matrix $\mathbf{I}$. Its objective includes both invariance term and redundancy reduction term, where the latter term can avoid trivial solutions. Inspired by Barlow Twins, VICReg (Bardes et al., 2021) proposes a new regularization term named invariance-covariance on feature dimension, and achieves similar results to Barlow Twins. However, both Barlow Twins and VICReg require large contrastive dimensions and the covariance regularization requires $\mathcal{O}(d^2)$ complexity, which could be also regarded as requiring negative pairs on feature dimension (each negative pair is composed of features in different dimensions across images).

**Whitening transformation.** Whitening, or sphering, is a common pre-processing step to transform random variables to orthogonality. However, due to rotational freedom, there are infinitely many possible whitening procedures. There is a diverse range of sphering methods, e.g. principal component analysis (PCA) (Jégou & Chum, 2012), Cholesky matrix decomposition (Higham, 1990) and zero-phase component analysis (ZCA) (Bell & Sejnowski, 1997). Compared with other whitening methods, ZCA remains maximal correlation of original data, which we will clarify in Sec. 3.6.

## 3 THE PROPOSED ZERO-CL

In this section, we present the framework of Zero-CL in detail, starting with the whitening transformation procedure, followed by the model's framework as well as the objective function. At last, we provide theoretical analysis on whitening from *maximal correlation* perspective.

### 3.1 WHITENING TRANSFORMATION

Given a set of images $\{\mathbf{x}\}_{i=1}^N$, we extract an embedding in contrastive space $\mathbf{z}_i = g(f(\mathbf{x}_i, \theta), \gamma)$ using an encoder network (He et al., 2016) $f(\cdot, \theta)$ and a MLP module $g(\cdot, \gamma)$, where $\mathbf{z}_i \in \mathbb{R}^{1 \times d}$. Denote the set of embeddings as $\mathbf{Z} \in \mathbb{R}^{N \times d}$, whitening transformation can be formulated as:

$$\mathbf{H} = (\mathbf{h}_1, \mathbf{h}_2, \cdots, \mathbf{h}_N) = \mathbf{W}\mathbf{Z}^\top \tag{1}$$

where $\mathbf{H} \in \mathbb{R}^{d \times N}$ is the whitened embedding matrix. The square matrix $\mathbf{W} \in \mathbb{R}^{d \times d}$ is the so-called whitening matrix. Since $var(\mathbf{H}) = \mathbf{I}$, it follows that $\mathbf{W}\mathbf{\Sigma}\mathbf{W} = \mathbf{I}$ and thus $\mathbf{W}(\mathbf{\Sigma}\mathbf{W}^\top\mathbf{W}) = \mathbf{W}$, which is fulfilled if $\mathbf{W}$ satisfies $\mathbf{W}^\top\mathbf{W} = \mathbf{\Sigma}^{-1}$. However, the constraint does not uniquely determine the whitening matrix $\mathbf{W}$. Specifically, for any $\mathbf{W}^{rot} = \mathbf{R}\mathbf{W}$ with orthogonal matrix, $\mathbf{R}$ will also be a whitening transformation matrix. Here we present ZCA whitening matrix in detail.

**ZCA-based** whitening takes $\mathbf{E}$ (stacks together eigenvectors of the covariance matrix) as the orthogonal matrix $\mathbf{R}$. Then ZCA whitening matrix can be written as:

$$\mathbf{W}^{ZCA} = \mathbf{E}\mathbf{\Lambda}^{-1/2}\mathbf{E}^\top \tag{2}$$

By rotation matrix $\mathbf{E}$, the whitened data will be as close as possible to original data (see Theorem 1). Note that in low-rank conditions, i.e., some eigenvalues equal to 0, a common approach is replacing $\mathbf{\Lambda}^{-1/2}$ with $(\mathbf{\Lambda} + \lambda\mathbf{I})^{-1/2}$ to prevent zero division, where $\lambda = 1e-4$ as default.

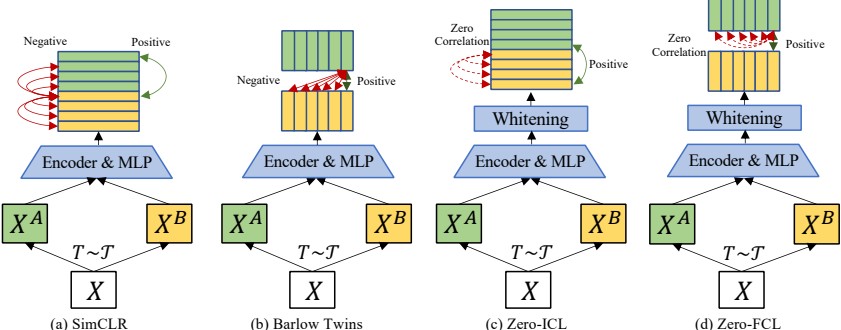

Figure 2: Frameworks of SimCLR (a), Barlow Twins (b), Zero-ICL (c) and Zero-FCL (d), where row vector means one embedding of instance, and column vector means vector composed by $i$-th dimension feature values among all instance. The solid red line means negative pairs and the corresponding objective function includes such negative term. The dash red lines mean the two vectors are orthogonal and thus, there is no need for negative term in objectives.

## 3.2 SYMMETRIC OBJECTIVE FUNCTION BACKGROUND

For instance-wise contrastive learning, to avoid trivial solutions in the symmetric framework, the objective function usually can be divided into alignment and uniformity terms (Arora et al., 2019). We now review the loss formulated in SimCLR (Chen et al., 2020) as:

$$\mathcal{L}_{info} = \underbrace{\mathbb{E}_{(x,x^+)\sim p_{pos}}[-f(x)^\top f(x^+)/\tau]}_{\text{alignment}} + \underbrace{\mathbb{E}_{\substack{(x,x^+)\sim p_{pos}\\ \{x_i^-\}_{i=1}^N\sim p_{data}}}\left[\log(e^{f(x)^\top f(x^+)/\tau} + \sum_i e^{f(x)^\top f(x_i^-)/\tau})\right]}_{\text{uniformity}}$$

(3)

where $\tau$ is the temperature hyper-parameter. $x^+$ and $x^-$ are positive and negative samples, respectively. The first term is the so-called alignment, which maximizes the similarity between positive pairs. The second term is the key to prevent trivial solution, pushing all the data points on a unit hyper-sphere uniformly. Then, we rewrite the feature-wise objective of Barlow Twins as:

$$\mathbf{L}_{BT} = \underbrace{\sum_i (1 - \mathbf{C}_{ii})^2}_{\text{invariance}} + \lambda \underbrace{\sum_i \sum_{j\neq i} \mathbf{C}_{ij}^2}_{\text{redundancy reduction}}$$

(4)

where $\mathbf{C} \in \mathbb{R}^{d\times d}$ is the covariance matrix. The first term encourages correlation of the same feature across different views, while the second term minimizes correlation of different features. We can observe that the objective of both instance-wise and feature-wise methods requires square order calculation complexity, where the first term (alignment, invariance) requires linear order complexity and the second term (uniformity, redundancy reduction) requires square order complexity. We wonder if we can discard the second term, i.e., completely discard negative pairs.

## 3.3 INSTANCE-WISE AND FEATURE-WISE WHITENING

Inspired by W-MSE (Ermolov et al., 2021), we propose two efficient methods in SSL, which are named Zero-ICL (instance-wise) and Zero-FCL (feature-wise). Zero-ICL can replace the uniformity term in Eq. 3, while Zero-FCL can replace the redundancy reduction term in Eq. 4.

**Instance-wise whitening.** Given a set of centered embeddings $\mathbf{Z} \in \mathbb{R}^{N\times d}$, different from traditional whitening methods, we calculate the **affinity** matrix by $\mathbf{S} = \mathbf{Z}\mathbf{Z}^\top$, where $\mathbf{S} \in \mathbb{R}^{N\times N}$. Then the ZCA whitening matrix $\mathbf{W}^{I-ZCA}$ can be calculated by:

$$\mathbf{W}^{I-ZCA} = \mathbf{E}\mathbf{\Lambda}_{\mathbf{S}}^{-1/2}\mathbf{E}^\top$$

(5)

where $\mathbf{E} \in \mathbb{R}^{N\times N}$ stacks eigenvector of the affinity matrix and $\mathbf{\Lambda}_{\mathbf{S}}$ is the diagonal variance matrix. One question is how this transformation replaces the uniformity term. We denote the whitened embedding matrix as $\mathbf{H}$. For a rotation orthogonal matrix $\mathbf{E}$ i.e., $\mathbf{E}^\top\mathbf{E} = \mathbf{E}\mathbf{E}^\top = \mathbf{I}$, the following

formula becomes $\mathbf{I} \in \mathbb{R}^{N \times N}$. It means embeddings of different samples are strictly orthogonal.

$$\mathbb{E}\left[\mathbf{H}\mathbf{H}^{\top}\right] = \mathbb{E}\left[(\mathbf{E}\boldsymbol{\Lambda}_{\mathbf{S}}^{-1/2}\mathbf{E}^{\top}\mathbf{Z})(\mathbf{E}\boldsymbol{\Lambda}_{\mathbf{S}}^{-1/2}\mathbf{E}^{\top}\mathbf{Z})^{\top}\right] \tag{6}$$

Then, we can replace the second part in InfoNCE with instance-wise whitening transformation.

**Feature-wise whitening.** Similar to instance-wise whitening transformation, it is calculated by:

$$\mathbf{W}^{F-ZCA} = \mathbf{E}\boldsymbol{\Lambda}_{\mathbf{C}}^{-1/2}\mathbf{E}^{\top} \tag{7}$$

where $\mathbf{C}$ is the **covariance** matrix, $\boldsymbol{\Lambda}_{\mathbf{C}}$ and $\mathbf{E} \in \mathbb{R}^{d \times d}$ are the diagonal variance matrix and eigen-matrix of $\mathbf{C}$, respectively. Denote whitened embedding matrix as $\mathbf{H} \in \mathbb{R}^{N \times d}$, we have $\mathbb{E}[\mathbf{H}^{\top}\mathbf{H}] = \mathbf{I}$, i.e., the correlation of the different features is equal to 0. Hence, redundancy reduction term in Eq. 4 can be replaced with feature-wise whitening transformation.

### 3.4 FRAMEWORK AND OBJECTIVE FUNCTION

Like other methods in SSL (Chen et al., 2020; Zbontar et al., 2021; Chen & He, 2021), given a batch $\{\mathbf{X}_i\}_{i=1}^{K}$ sampled from a dataset, we first augment data via a distribution of data augmentation $\mathcal{T}$. We term the two views as $\mathbf{X}^A$ and $\mathbf{X}^B$. The two batches are then fed to an encoder (He et al., 2016; Kipf & Welling, 2016) $f(\cdot, \theta)$, producing batches of embeddings $\mathbf{T}^A$ and $\mathbf{T}^B$. Then, a projector module $g(\cdot, \gamma)$ is used to map representation to contrastive space. The two batches of embeddings are then fed into projector $g$, producing $\mathbf{Z}^A \in \mathbb{R}^{K \times d}$ and $\mathbf{Z}^B \in \mathbb{R}^{K \times d}$. Since whitening transformation requires centered vector, we use standard scaling method to each batch of embeddings. Note that to calculate **instance-wise** and **feature-wise** whitening transformation matrix, we must center embeddings on **feature** dimension and **instance** dimension, respectively. We denote feature-wise and instance-wise centered embeddings as $\mathbf{Z}^{Fea}$ and $\mathbf{Z}^{Ins}$ as follow:

$$\mathbf{Z}_{i,\cdot}^{A(B),Ins} = \frac{\mathbf{Z}_i^{A(B)} - \mu_i}{\sqrt{\sigma_i^2 + \epsilon}} \; 1 \le i \le d, \quad \mathbf{Z}_{\cdot,i}^{A(B),Fea} = \frac{\mathbf{Z}_i^{A(B)} - \mu_i}{\sqrt{\sigma_i^2 + \epsilon}} \; 1 \le i \le K \tag{8}$$

where $\mathbf{Z}_{i,\cdot}$ is the $i$-th vector in $\mathbf{Z}$ and $\mathbf{Z}_{\cdot,i}$ is the vector composed of each value at dimension $i$ in all embeddings in $\mathbf{Z}$. $\mu$ and $\sigma$ are mean and variance of one vector. $\epsilon$ is set 1e-4 as default.

**Instance-wise objective.** Now we have two batches of instance-wise centered embeddings $\mathbf{Z}^{A,Ins}$ and $\mathbf{Z}^{B,Ins}$. By Eq. 5, we can obtain instance-wise whitened embeddings $\mathbf{H}^{A,Ins}$ and $\mathbf{H}^{B,Ins}$. Then, the instance-wise objective can be formulated as:

$$\mathcal{L}_{Ins} = \sum_{i}^{K} \left(1 - \sum_{d} \mathbf{H}_{i,d}^{A,Ins} \cdot \mathbf{H}_{i,d}^{B,Ins}\right)^2 \tag{9}$$

where $\mathbf{H}_{i,d}$ represents the $d$-th feature value of the $i$-th instance.

**Feature-wise objective.** Similar to Eq. 9, we can obtain feature-wise whitened embeddings $\mathbf{H}^{A,C-ZCA}$ and $\mathbf{H}^{B,C-ZCA}$ via Eq. 7. Then the feature-wise objective can be formulated as:

$$\mathcal{L}_{Fea} = \sum_{d} \left(1 - \sum_{i}^{K} \mathbf{H}_{i,d}^{A,Fea} \cdot \mathbf{H}_{i,d}^{B,Fea}\right)^2 \tag{10}$$

where $K$ denotes the batch size. The overall loss can be formulated as:

$$\min_{\theta,\gamma} \mathcal{J} = \mathcal{L}_{Ins} + \lambda \cdot \mathcal{L}_{Fea} \tag{11}$$

We set the hyper-parameter $\lambda = 1$ in this paper. Note that both two losses can work independently and we term $\mathcal{L}_{Ins}$ only as Zero-ICL (instance-wise) and $\mathcal{L}_{Fea}$ only as Zero-FCL (feature-wise).

### 3.5 EMPIRICAL ANALYSIS

Here for simplicity, we discuss feature-wise whitening. From the redundancy reduction perspective, high-quality self-supervised embeddings require that: 1) embeddings are in a non-trivial constant distribution, where trivial distribution means all the embeddings collapse to a single point; 2) positive image pairs share similar semantics; and 3) correlation of the different feature should be zero. Considering feature-wise SSL, for simplicity, denote $i$-th feature in $\mathbf{Z}$ as $\mathbf{Z}_{\cdot,i}$, where $\mathbf{Z} \in \mathbb{R}^{N \times d}$ and $\mathbf{Z}_{\cdot,i} \in \mathbb{R}^{N \times 1}$, we can formulate this problem as follows:

$$\min_{\theta,\gamma} \mathbb{E}\left[dist(\mathbf{Z}_{\cdot,i}^A, \mathbf{Z}_{\cdot,i}^B)\right] \;\; s.t. \;\; cov(\mathbf{Z}^A) = cov(\mathbf{Z}^B) = \mathbf{I} \tag{12}$$

where $dist(\cdot, \cdot)$ can be quantified as any similarity metric. To solve the conditional optimization problem, previous works (Zhang et al., 2021a; Zbontar et al., 2021) transform the hard condition to soft condition via the Lagrangian multiplier. The objective becomes:

$$\min_{\theta,\gamma} \mathbb{E}\left[dist(\mathbf{Z}^A_{\cdot,i}, \mathbf{Z}^B_{\cdot,i})\right] + \lambda \cdot \left(\|(\mathbf{Z}^A)^\top \mathbf{Z}^A - \mathbf{I}\|^2 + \|(\mathbf{Z}^B)^\top \mathbf{Z}^B - \mathbf{I}\|^2\right) \tag{13}$$

However, the second term of the above objective function requires square order complexity. Hence, we propose another approach to solve this hard conditional optimization problem, i.e., whitening transform the representations before calculating loss. Then the objective of Eq. 12 is equal to

$$\min_{\theta,\gamma} \mathbb{E}\left[dist(\mathbf{H}^A_{\cdot,i}, \mathbf{H}^B_{\cdot,i})\right] \tag{14}$$

where $\mathbf{H}$ is the whitened embeddings. Finally, our objective only uses $\mathcal{O}(d)$ complexity to solve the conditional problem, which is similar to the instance-wise branch.

### 3.6 THEORETICAL ANALYSIS ON WHITENING

By Eq. 1, for any $\mathbf{W}^{rot} = \mathbf{R}\mathbf{W}$, which satisfies $\mathbf{R}^\top \mathbf{R} = \mathbf{I}$, $\mathbf{W}^{rot}$ can be a whitening matrix. For SSL, whitening transformation aims to make different instances or features orthogonal, and also remains as much original information as possible. The problem can be formulated as:

$$\min_{\mathbf{R}} \mathbb{E}[(\mathbf{H} - \mathbf{Z})^\top (\mathbf{H} - \mathbf{Z})]\ \ s.t.\ \mathbf{H}^\top \mathbf{H} = \mathbf{I},\ \mathbf{R}^\top \mathbf{R} = \mathbf{I},\ \mathbf{W}^\top \mathbf{W} = \mathbf{\Sigma}^{-1} \tag{15}$$

where $\mathbf{R}$, $\mathbf{\Sigma}$ are the rotation matrix and covariance matrix. $\mathbf{Z}$ and $\mathbf{H}$ are embeddings before and after whitening, respectively. Since $\mathbf{H}$ and $\mathbf{Z}$ are two centered scaling matrices, we can rewrite the constraint as follows, and thus minimizing the variance between whitened data and original data equals to maximizing the second term:

$$\mathbb{E}\left[(\mathbf{H} - \mathbf{Z})^\top (\mathbf{H} - \mathbf{Z})\right] = 2\mathbf{I} - 2\text{tr}\left(\mathbf{R}\mathbf{E}\mathbf{\Lambda}^{-1/2}\mathbf{E}^\top\right) \tag{16}$$

**Theorem 1** *Maximization of $tr(\mathbf{R}\mathbf{E}\mathbf{\Lambda}^{-1/2}\mathbf{E}^\top)$ uniquely determines the rotation matrix $\mathbf{R} = \mathbf{I}$.*

**Proof 1** *Since $\mathbf{\Lambda}$ is a diagonal matrix, we can write Eq. 16 as:*

$$tr(\mathbf{R}\mathbf{E}\mathbf{\Lambda}^{-1/2}\mathbf{E}^\top) = tr(\mathbf{\Lambda}^{-1/2}\mathbf{E}^\top \mathbf{R}\mathbf{E}) = \sum_i \mathbf{\Lambda}_{ii}^{-1/2}\mathbf{A}_{ii} \tag{17}$$

*where $\mathbf{A} = \mathbf{E}^\top \mathbf{R}\mathbf{E}$. Since $\mathbf{R}$ and $\mathbf{E}$ are both orthogonal, $\mathbf{A}$ is also an orthogonal matrix. So we have $\mathbf{A}_{ii} \leq 1$. Note that $\mathbf{E}$ is a rotation matrix, i.e., $\mathbf{E}^\top \mathbf{E} = \mathbf{I}$. Thus, if and only if $\mathbf{R} = \mathbf{I}$, we can obtain the maximum of Eq. 17. Then, we can complete the proof.* □

## 4 EXPERIMENTS

We use the following datasets. The implementation and protocol details are given in Appendix.

1) Two small-scale datasets CIFAR-10, CIFAR-100 (Krizhevsky et al., 2009), are composed of $32 \times 32$ images with 10 and 100 classes.

2) ImageNet-100 and ImageNet-1k (Deng et al., 2009) sets include 100 and 1k classes, respectively. The datasets are well-balanced in class distribution and the images contain an iconic view of objects, which are the commonly used benchmark in SSL (Zbontar et al., 2021).

Table 1: Accuracy on ImageNet with ResNet-50.

| Method | 100 eps | | 400 eps | |
|---|---|---|---|---|
| | acc@1 | acc@5 | acc@1 | acc@5 |
| SimCLR | 66.5 | 86.31 | 69.2 | 89.0 |
| Moco v2 | 67.1 | 87.59 | 71.1 | 90.1 |
| BYOL | 66.3 | 87.66 | **73.2** | **91.3** |
| SwAV | 66.5 | 87.71 | 70.7 | ∼ |
| SimSiam | 68.1 | 88.48 | 70.8 | ∼ |
| Barlow Twins | 67.7 | 88.36 | 73.1 | 91.0 |
| Zero-CL | **68.9** | **88.74** | 72.6 | 90.5 |

**Main comparison.** We mainly conduct experiments on CIFAR-10/100 and ImageNet-100/1k datasets. For fair comparison, we set batch size as 128 on ImageNet-100, 1024 on ImageNet-1k, and 256 on CIFAR-10/100. Table 1 and 2 show the best results on different datasets across different methods, where our method outperforms most of the prior arts with symmetric architecture and less complexity. We modify our code based on Barlow Twins, and the reported results are strictly followed official code. Note that on CIFAR-10 and CIFAR-100 datasets, we remove the first maxpool

Table 2: Main comparison on CIFAR and ImageNet-100. Proj. and Pred. mean the hidden dimension in projector and predictor. Negs means negatives (both feature-wise and instance-wise). All methods are trained 1000 epochs on CIFAR-10/100 (batch size 256) and 400 epochs on ImageNet-100 (batch size 128). $N$ means number of samples, $d$ means hidden dimensions, $M$ is the number of views and $C$ is the clustering classes. The complexity only considers objective function in line with peer works e.g. (Zhang et al., 2021a), since other parts like feedforward computing share the same overhead. Note that some results are directly quoted from solo-learn (da Costa et al., 2021).

| | Method | Proj. | Pred. | Negs. | Complexity | CIFAR-10 Acc@1 | CIFAR-10 Acc@5 | CIFAR-100 Acc@1 | CIFAR-100 Acc@5 | ImageNet-100 Acc@1 | ImageNet-100 Acc@5 |
|---|---|---|---|---|---|---|---|---|---|---|---|
| Asymmetric | BYOL | 4096 | 8192 | ✗ | $\mathcal{O}(N)$ | 92.61 | **99.82** | **70.18** | 91.36 | **80.09** | 94.99 |
| | DINO | 2048 | ✗ | ✗ | $\mathcal{O}(N)$ | 89.19 | 99.31 | 66.38 | 90.18 | 74.84 | 92.92 |
| | SimSiam | 2048 | 512 | ✗ | $\mathcal{O}(N)$ | 90.51 | 99.72 | 65.86 | 89.48 | 77.04 | 94.02 |
| | MoCo V2 | 2048 | ✗ | ✔ | $\mathcal{O}(NK)$ | **92.94** | 99.79 | 69.54 | **91.49** | 78.2 | **95.5** |
| | ReSSL | 2048 | ✗ | ✔ | $\mathcal{O}(N^2)$ | 90.63 | 99.62 | 65.83 | 89.51 | 76.59 | 94.41 |
| Symmetric | VICReg | 2048 | ✗ | ✔ | $\mathcal{O}(N + d^2)$ | 90.07 | 99.71 | 68.54 | 90.83 | 79.22 | **95.06** |
| | SwAV | 2048 | ✗ | ✔ | $\mathcal{O}(NC)$ | 89.17 | 99.68 | 64.67 | 88.52 | 74.28 | 92.84 |
| | W-MSE | 256 | ✗ | ✗ | $\mathcal{O}(NM^2)$ | 88.18 | 99.61 | 61.29 | 87.11 | 69.06 | 91.22 |
| | SimCLR | 2048 | ✗ | ✔ | $\mathcal{O}(N^2)$ | 90.74 | 99.75 | 65.39 | 88.58 | 77.48 | 93.42 |
| | Barlow Twins | 256 | ✗ | ✔ | $\mathcal{O}(d^2)$ | 87.39 | 99.42 | 57.92 | 85.23 | 67.21 | 90.64 |
| | Barlow Twins | 2048 | ✗ | ✔ | $\mathcal{O}(d^2)$ | 89.57 | 99.73 | 69.18 | 91.19 | 78.62 | 94.72 |
| | Zero-FCL | 256 | ✗ | ✗ | $\mathcal{O}(d)$ | 89.77 | 99.73 | 66.81 | 89.71 | 75.67 | 93.63 |
| | Zero-FCL | 2048 | ✗ | ✗ | $\mathcal{O}(d)$ | 90.51 | 99.76 | 70.25 | 91.96 | **79.32** | 94.94 |
| | Zero-ICL | 256 | ✗ | ✗ | $\mathcal{O}(N)$ | 90.47 | 99.76 | 69.33 | 91.62 | 78.02 | 95.61 |
| | Zero-CL | 2048 | ✗ | ✗ | $\mathcal{O}(N + d)$ | **90.81** | **99.77** | **70.33** | **92.05** | 79.26 | 94.98 |

Table 3: Convergence rate. Cholesky-FCL and Cholesky-ICL mean replace ZCA whitening with Cholesky whitening on feature and instance-wise, respectively. For fairness, we set the dimension of projection head 128-128-128 and batch size 256.

| Method | 100 eps Acc@1 | 100 eps Acc@5 | 1000 eps Acc@1 | 1000 eps Acc@5 |
|---|---|---|---|---|
| Barlow Twins | 80.65 | 99.10 | 87.27 | 99.46 |
| SimCLR | 78.39 | 98.84 | 90.16 | 99.66 |
| Zero-ICL | 84.59 | 99.28 | 90.15 | **99.71** |
| Cholesky-ICL | 76.51 | 98.75 | 85.24 | 99.16 |
| Zero-FCL | 83.2 | 99.11 | 88.25 | 99.41 |
| Cholesky-FCL | 82.83 | 99.02 | 87.63 | 99.37 |
| Zero-CL | **85.01** | **99.36** | **90.24** | 99.70 |

Table 4: Analysis on negatives. The hidden dimension of Zero-ICL and -FCL is set 256 and 2048. SimCLR w/o negatives means directly using MSE loss after $l_2$ normalization between positive embeddings, while the reproduced Barlow Twins w/o negatives means only using the invariance term in Eq. 4.

| Method | w/o negatives Acc@1 | w/o negatives Acc@5 | w/ negatives Acc@1 | w/ negatives Acc@5 |
|---|---|---|---|---|
| Barlow Twins | 16.98 | 41.26 | 65.81 | 90.18 |
| SimCLR | 6.92 | 22.36 | 64.98 | 89.91 |
| Zero-ICL | **67.54** | **90.85** | 62.61 | 87.89 |
| Zero-FCL | 67.31 | 90.76 | 65.95 | 90.15 |

layer and modify the first convolutional layer with kernel size 3 and strides 1 in ResNet-18, which are commonly used tricks in low-resolution datasets (Chen et al., 2020). The MLP (Projector) in our methods is quantified as three linear layers with two BNs (Ioffe & Szegedy, 2015) and ReLU activation function. Table 2 shows the results on CIFAR and ImageNet-100 datasets. We compare our Zero-FCL with Barlow Twins with the same hidden dimension. With 256 hidden dimension, Zero-FCL outperforms Barlow Twins by **8.89%** and **8.46%** in top-1 accuracy on CIFAR-100 and ImageNet-100, respectively. For hidden dimension 2048, Zero-FCL outperforms Barlow Twins 1.07% and 0.7% top-1 accuracy on CIFAR-100 and ImageNet-100 datasets. Note that although Zero-ICL is an instance-wise method and the batch size is set only 128 or 256, Zero-ICL still gets 69.33% top-1 accuracy on CIFAR-100, which is **3.94%** higher than SimCLR.

**Convergence rate.** Many experiments (Gutmann & Hyvärinen, 2010; Chen et al., 2020) have shown that contrasting on high dimension sphere converge faster than low dimension sphere. Hence, to explore if our methods can converge fast on a low contrastive dimension, we conduct extensive experiments with 100 and 1000 epochs on CIFAR-10 dataset. For fair comparisons, we choose two symmetric methods, i.e., SimCLR (instance-wise) and Barlow Twins (feature-wise) as baselines. Table 3 illustrates our method outperforms the mentioned two methods. Specifically, Zero-FCL outperforms Barlow Twins and SimCLR by a large range (**4.36%**, **6.63%** top-1 accuracy) with 100 epochs, while the improvement decreases after 1000 epochs. We conjecture the reason may be that

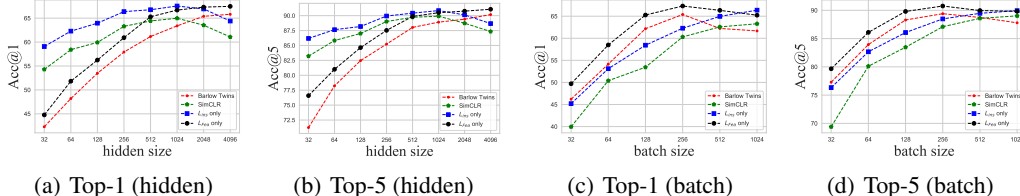

| (a) Top-1 (hidden) | (b) Top-5 (hidden) | (c) Top-1 (batch) | (d) Top-5 (batch) |

Figure 3: Classification accuracy on CIFAR-100 under different hidden dimensions in (a), (b) and batch size in (c), (d). For (a) and (b), the batch size of Zero-ICL and Zero-FCL are set 256 and 1024. For (c) and (d), the hidden dimension of Zero-ICL and Zero-FCL is set 256 and 2048, respectively.

Barlow Twins and SimCLR have two terms to optimize, where the first term is the key to learn good embeddings (Arora et al., 2019), and the second term (uniformity or redundancy reduction) is the key to prevent trivial solutions. However, the second term is hard to converge (Wang & Isola, 2020). In contrast, our methods exactly discard the second term via whitening transformation. Thus, our method converges at a faster speed. We also report the results of different whitening transformation methods, i.e., Cholesky and ZCA. We find that for instance-wise contrastive learning, Cholesky-based decomposition gets much lower accuracy scores rather than the ZCA-based method. For feature-wise contrastive learning, Cholesky gets a slightly lower accuracy than ZCA-based. We conjecture this is because ZCA-based whitening remains maximal information of original data (see Theorem 1), which is important for instance-wise learning (slight rotation on instance dimension will heavily influence linear evaluation), while for feature-wise learning, when the features are permuted or rotated, it may not cause serious influence for linear evaluation (Mitra et al., 2002).

**Sensitivities to batch size.** To explore our method's sensitiveness to batch size, we conduct experiments with different batch sizes from $32 \sim 1024$. For a fair comparison, each method with a different batch size is trained with 50K **iterations**. Fig. 3(c) shows the results on CIFAR-100. We can observe SimCLR and Zero-ICL get lower accuracy when batch size is small and this phenomenon is not surprising since instance-wise contrastive learning methods usually require a large batch size to create negative pairs (SimCLR) or exact statistics (Zero-ICL). Another observation is the accuracy of Zero-FCL saturates when $BatchSize = 256$, which is consistent with the results in (Zbontar et al., 2021). Note that under 1024 batch size, Zero-ICL outperforms SimCLR by about 3.1% top-1 accuracy and Zero-FCL outperforms Barlow Twins 3.8% top-1 accuracy. We guess that's because of the diagonal property of whitened embeddings, which makes our methods robust to batch size.

**Sensitivities to hidden dimensions.** We also explore the sensitivity of our method to hidden dimensions, which are set from $32 \sim 4096$. Note that the three linear layers in projection MLP $g$ are set as the same input and output dimension. For feature-wise methods (Barlow Twins, Zero-FCL), the batch size is set as 256 (best hyper-parameter in Fig. 3(c)). For instance-wise methods (SimCLR, Zero-ICL), we set batch size as 1024. All the results are get by 50K iterations. Fig. 3(a) shows classification accuracy with different hidden dimensions. For Barlow Twins and Zero-FCL, both two methods are heavily influenced by hidden dimensions and Zero-FCL outperforms Barlow Twins under all hidden dimensions settings. Oppositely, because the objective functions of SimCLR and Zero-ICL are instance wise, making them more robust to hidden dimensions than Barlow Twins.

**Ablation to negatives.** To show Zero-CL is indeed a negative-free method, we conduct an ablation study w/ and w/o negative samples and compare our methods with SimCLR, Barlow Twins. For our methods w/ negatives, we add the off-diag entry of cross-correlation matrix in our objective function, which is followed Barlow Twins (Zbontar et al., 2021). Table 4 shows results of four symmetric methods w/ and w/o negatives pairs. Performance of Barlow Twins significantly decreases and SimCLR directly crashes w/o negative pairs. Zero-FCL and Zero-ICL do not require negatives and perform the best without negatives. After adding negatives, both Zero-ICL and Zero-FCL lose some accuracy. We analyze that is because whitening transformation can be regarded as a constraint, while using negatives in the objective function is also a constraint. Such two constraints will trade off the weight of alignment part (Arora et al., 2019), where the alignment part is the key to learn representations in contrastive learning (Grill et al., 2020; Wang & Isola, 2020). Thus, the accuracy of Zero-ICL and Zero-FCL with negatives is lower than those without negatives.

**Breaking Symmetry.** In line with Barlow Twins (Zbontar et al., 2021), we conduct extensive experiments with different asymmetric methods, i.e., stop-gradient and predictor. For Zero-ICL, the objective only has an alignment part, which has been shown easily to collapse without the asym-

metric framework (Grill et al., 2020; Chen & He, 2021), while the proposed instance whitening can solve this problem well. Table 5 gives classification accuracy on ImageNet-100 with ResNet-18 backbone. In our experiments, the predictor module is composed of two linear layers with batch normalization. Similar to Barlow Twins, we find these asymmetries slightly decrease the performance of our network. For stop gradient, it is easy to understand the accuracy dropping, i.e., without considering collapse, the information provided by two views will be more precise than single view (Caron et al., 2020; Ermolov et al., 2021), while for predictor, we guess that's because overfitting by the non-linear transformation power of deep MLP (Goodfellow et al., 2016).

## 5 FURTHER DISCUSSION

We further compare our work with existing decorrelation methods and highlight our differences.

**Relation to CCA-SSG (Zhang et al., 2021a).** Inspired by classical Canonical Correlation Analysis, CCA-SSG proposes a new loss function for graph data learning, which is similar to Barlow Twins.

$$\mathcal{L}_{CCA} = \underbrace{\|\mathbf{Z}^A - \mathbf{Z}^B\|_2}_{\text{invariance}} + \lambda \left( \underbrace{\|(\mathbf{Z}^A)^\top \mathbf{Z}^A - \mathbf{I}\|_2 + \|(\mathbf{Z}^B)^\top \mathbf{Z}^B - \mathbf{I}\|_2}_{\text{decorrelation}} \right) \tag{18}$$

where the first term is similar to $\mathcal{L}_{BT}$. However, the decorrelation term of $\mathcal{L}_{CCA}$ aims to minimize the **intra**-correlation of different features, while the redundancy term of $\mathcal{L}_{BT}$ minimizes **cross**-correlation of different features. Our feature-wise whitening with $\mathcal{L}_{Fea}$ only requires the invariance term of CCA-SSG or Barlow Twins, while the other term of $\mathcal{L}_{CCA}$ is replaced by intra-feature-wise whitening, i.e., whitening transformation is conducted inside each batch view.

Table 5: Ablation study of asymmetric/symmetric architectures as generated by applying stop gradient ('SG')/predictor ('Pred') on one of the branches, by following (Zbontar et al., 2021).

| Method | SG | Pred. | Top-1 | Top-5 |
|---|---|---|---|---|
| Zero-FCL | ✗ | ✗ | **79.32** | **94.94** |
| | ✗ | ✔ | 76.57 | 94.18 |
| | ✔ | ✗ | 77.81 | 94.22 |
| | ✔ | ✔ | 74.81 | 92.81 |
| Zero-ICL | ✗ | ✗ | **78.02** | **94.11** |
| | ✗ | ✔ | 75.12 | 93.18 |
| | ✔ | ✗ | 74.87 | 92.72 |
| | ✔ | ✔ | 72.19 | 91.95 |

**Relation to W-MSE (Ermolov et al., 2021).** W-MSE firstly combines classical whitening transformation with contrastive learning. Compared with this whitening method, Zero-CL mainly has three differences. **1) Method.** W-MSE directly borrows whitening transformation on feature-wise, which is widely used in other domains (Huang et al., 2018; Zhang et al., 2021b). However, after feature-wise whitening, the instance-wise objective function is adopted, which is groundless and also limits their performance. Depart from W-MSE, Zero-CL aims to replace the second term of InfoNCE and $\mathcal{L}_{BT}$. Thus, Zero-ICL uses instance-wise objective function with instance-wise whitening, and Zero-FCL uses feature-wise objective function with feature-wise whitening, which is meaningful and more comprehensible. Such methods also leverage the performance (See Table 2). **2) Technology.** Instead of directly using Cholesky or PCA-based whitening, we first analyze which whitening transformation method is we really need. **3) SSL settings.** W-MSE requires multiple views to boost the accuracy, while Zero-CL can get higher accuracy with only two views.

## 6 CONCLUSION

We have proposed two new methods to prevent degenerate solutions in the symmetric architecture of SSL. We first analyze existing objective functions in SSL ($\mathcal{L}_{Info}$, $\mathcal{L}_{BT}$ and $\mathcal{L}_{CCA}$). Then, we show why Zero-FCL and Zero-ICL can replace the uniformity term of $\mathcal{L}_{Info}$ and redundancy term of $\mathcal{L}_{BT}$, respectively. We further give theoretical analysis on how ZCA based whitening remains maximal information rather than other whitening transformation methods. Finally, we conduct experiments on three image benchmarks and results show that our methods can outperform or be on par with prior arts. Further, to the best of our knowledge, Zero-ICL is the first attempt for instance-wise whitening and we hope it can bring more inspiration to other tasks.

### ACKNOWLEDGMENTS

This work was in part supported by National Key Research and Development Program of China (2020AAA0107600), Shanghai Municipal Science, Technology Major Project (2021SHZDZX0102) and SenseTime Collaborative Research Grant.

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

# A  APPENDIX

## A.1  IMPLEMENTATION IN VISION

**Image augmentations.** Each input image is transformed twice to generate two different views mentioned before. The image augmentation pipeline follows Barlow Twins (Zbontar et al., 2021), composed of: random cropping, resizing to $224 \times 224$ ($32 \times 32$ for CIFAR), horizontal flipping, color jittering, converting to gray-scale, Gaussian blurring, and solarization. The last five are applied randomly on two views with different probabilities.

**Architecture.** Followed by recent works (Chen et al., 2020; Zbontar et al., 2021), the encoder consists of ResNet-50 and ResNet-18 (He et al., 2016) (without the final classification layer, 2048, 512 output units, respectively) followed by a MLP module. In line with Barlow Twins (Zbontar et al., 2021), the MLP module contains three linear layers, each with the same output units. The first two layers of the MLP are followed by a BatchNorm layer (Ioffe & Szegedy, 2015).

**Optimization.** Similar to previous works (Grill et al., 2020; Zbontar et al., 2021), we use the LARS optimizer (You et al., 2017) on both three image datasets. We use a learning rate of 0.2 for the weights and 0.005 for the biases and batch normalization parameters. We multiply the learning rate by batch size and divide it by 256. We use a learning rate warm-up period of the first 10 epochs, after which we reduce the learning rate by a factor of 1000 using a cosine decay scheduler (Loshchilov & Hutter, 2016). For CIFAR-10 and CIFAR-100, we use single 1080 GPU. For ImageNet-100, the batch size is set as 128 as default, and we use 8 Tesla V100 16G GPUs. For ImageNet, we use 64 1080Ti GPUs.

**Evaluation.** We train a linear classifier on three vision datasets on top of fixed representations of ResNets pre-trained by Zero-CL. Specifically, the linear classifier is trained for 100 epochs with a learning rate of 0.3 and a cosine learning rate scheduler. We minimize the cross-entropy loss with SGD optimizer with momentum 0.9 and weight decay 1e-6. In line with previous arts (Zbontar et al., 2021; Chen et al., 2020), we set batch size 256. At the inference stage, we resize the image to 256 $\times$ 256 and center crop it to a size of $224 \times 224$.

