# OpenReview forum: "Zero-CL: Instance and Feature decorrelation for negative-free symmetric contrastive learning"
_ICLR.cc/2022/Conference — ICLR 2022 Poster_

### Official Review · Reviewer_1y87 · 2021-10-21

**Correctness:** 3
**Technical Novelty And Significance:** 3
**Empirical Novelty And Significance:** 3
**Recommendation:** 5
**Confidence:** 4

**Main Review:**

+ This paper dedicates to whitening transformation with a ZCA-based solution. It is intuitive to apply the whitening to both instance and feature dimensions, which makes the objective function of ICL and FCL consistent.
+ Paper is well presented and easy to follow.

- The key concerns came from the symmetric framework. Despite the fancy and consistent solution, the problem is down to the different properties of feature dimensions and instances. For image classification, we believe the ideal feature representation should have decorrelated non-redundant dimensions. However, the correlation between instances is inevitable - the inner-class correlation should be encouraged. Is a pure orthogonal matrix that pushes every instance unique the best solution? Should it be a "block-based" matrix with ones filled in the same class?
- Another big concern is about the claimed contribution in complexity. It is argued the second term in Eq(13) results in square order complexity. And the improvement is made by the replacement of the whitened embedding. I am not super clear why this can reduce the computational complexity.


- Eq (9) seems an error: should it be Z_{i,d}^{A,Ins}\dot Z_{i,d}^{B,Ins}?
- Decorrelating for redundancy removal can often result in low-rank effects (it is equivalent to implicit feature selection or sampling) in the transformed matrix. It would be good to discuss and potential properties and problems resulting from the low-rank issues.
- In the literature review, the background, properties and advantages of ZCA compared to PCA and Cholesky should be extended to clarify the motivation of using ZCA.
- Eq(4) the "1" should be in bold to denote an identity matrix?


**Summary Of The Paper:**

This paper addresses the collapse problem during self-supervised contrastive learning. ICL and FCL methods are proposed to decorrelate instances and features. Zero-Contrastive Learning can discard negative pairs with advantages in negative free, reducing correlation of different features to zero and retaining information during transformation. Promising experimental results are demonstrated on CIFAR and ImageNet.

**Summary Of The Review:**

Overall, it is a good paper. I am keen to understand why the symmetric framework can be consistently applied to features and instances given they have different properties in the contexts of image classification. I am also not clear about the complexity reduction, which seems to be claimed as a very important contribution in this paper.

Minor issues include some typos in equations and the discussion of low-rank property and ZCA background.

---

> ### Author Response · Authors · 2021-11-10
> **Response to Reviewer 1y87**
>
> Thanks for your suggestions. Due to some misunderstandings, we will give explanations in detail.
>
> **Key concerns of the symmetric framework.** From the paper [3], we know that the uniformity part and alignment part of InfoNCE [4]. However, it also brings some misunderstandings. Please take an in-depth consideration, if we push three $l_2$ normalized vectors in a unit circle (2-$d$) uniformly, the angle of each two vectors is equal to $120^{\circ}$. Then, for in 3-d space, there are maximal 6 vectors that can be uniformly distributed in the unit sphere with equal angles [5] and the angle is about $116.6^{\circ}$. Then, how about 256, 1024 even 4096 dimensions? For 3-d space, if we force pulling more than 6 vectors to the sphere uniformly, the angles between each two vectors are not equal, which is your main concerns (images with similar semantic information should share similar embeddings). Note that in our experiments, the dimension $d$ and batch size $N$ are in the same order of magnitude. Hence, even if uniform distribution (objective of InfoNCE), the angles between every two vectors are also equal to one constant. For Zero-ICL, the angle is $90^{\circ}$ (orthogonal), which indicates the reason why Zero-ICL can work well. However, we think the "block-based" idea is cool, which is worthy to explore in future work.
>
> **Complexity.** For instance-wise methods (similar for feature-wise), consider the relation of Zero-ICL and SimCLR. Given two set of embeddings (view A and view B) $Z^{A}=\{z_1^{A}, z_2^{A}, \cdots z_n^{A}\}$ and  $Z^{B}=\{z_1^{B}, z_2^{B}, \cdots, z_n^{B}\}$. The later one (SimCLR [1]) calculates the similarity ($z_i^{A}, z_j^B, 1 \leq i \leq N, 1 \leq j \leq N$) of each two embeddings of the two sets, which will takes $\mathcal{O}(N^2)$ time complexity. However, in our Zero-ICL, we first transform the two matrix via whitening to obtain $H^{A}=\{h_1^{A}, h_2^{A}, \cdots, h_n^{A}\}$ and $H^{B}=\{h_1^{B}, h_2^{B}, \cdots, h_n^{B}\}$. Then we only maximize the similarity of $h_i^{A}, h_i^{B}, 1\leq i\leq N$, which only takes $\mathcal{O}(N)$ complexity. One of the concerns is the complexity of whitening transformation. However, thanks to the work randomized SVD, we can calculate the whitening matrix with a very fast speed.
>
> **Typos in Eq. 9** Thanks for your careful reading. We have corrected the typo.
>
> **low-rank issue.** It's a classical problem, i.e., the input matrix is not full-rank or low-rank. One of the solution is modifying Eq. 7 to $W=E(\Lambda_{C}+\lambda I)^{-1/2}E^{\top}$.
>
> **Literature review.** We will add some discussion to PCA and Cholesky-based methods.
>
> **Question in Eq. 4.** The formulation is copied from Barlow Twins [1]. Actually, it should be 1, since $C_{ij}$ and $C_{ii}$ are numeric numbers but not vector or matrix.
>
> [1] Zbontar J, Jing L, Misra I, et al. Barlow twins: Self-supervised learning via redundancy reduction[J]. arXiv preprint arXiv:2103.03230, 2021.
>
> [2] Chen T, Kornblith S, Norouzi M, et al. A simple framework for contrastive learning of visual representations[C]//International conference on machine learning. PMLR, 2020: 1597-1607.
>
> [3] Wang T, Isola P. Understanding contrastive representation learning through alignment and uniformity on the hypersphere[C]//International Conference on Machine Learning. PMLR, 2020: 9929-9939.
>
> [4] Oord A, Li Y, Vinyals O. Representation learning with contrastive predictive coding[J]. arXiv preprint arXiv:1807.03748, 2018.
>
> [5] Jiang Z, Tidor J, Yao Y, et al. Equiangular lines with a fixed angle[J]. Annals of Mathematics, 2021, 194(3): 729-743.

---

> ### Author Response · Authors · 2021-11-30
> **Look forward to your further reply.**
>
> Dear reviewer, thanks again for your comments and valuable suggestions.
>
> Currently, we have made further clarification in the updated manuscript to address your concerns.
>
> **Instance-wise solutions:** we analyze in one mini-batch, the optimal solution is angles of each two embeddings equal to a constant (due to the given literature about equiangular lines in the previous reply). Considering InfoNCE, the solution in one mini-batch is also pulling the embeddings uniform (angles of each two embeddings are equivalent), which is consistent with our instance-wise whitening. The only difference is the angle, where Zero-ICL sets $90^{\circ}$, while the optimal solution equals one constant. For further clarify, please re-read our previous response.
>
> **Complexity:** we argue the linear-order complexity in the objective function since both Zero-ICL and Zero-FCL do not require pair-wise similarity calculations. The analysis and wall-clock time are given in reply to Reviewer kq6q and AaoY, respectively.
>
> Then, we also add the discussion to PCA-based method and low-rank effects in our modified manuscripts. We sincerely look forward to your reply and we could provide more information if needed.

---

### Official Review · Reviewer_kq6q · 2021-11-02

**Correctness:** 3
**Technical Novelty And Significance:** 3
**Empirical Novelty And Significance:** 3
**Recommendation:** 6
**Confidence:** 4

**Main Review:**

This loss function is somewhat new. However, the justification is relatively weak. The work is good, though at this stage, the authors have only been able to show it gives satisfactory results. It is unclear whether this is just an incremental work among the recent advances.

This formulation is still using other samples as negative samples. The negative samples are unrelated to the positive samples  in the loss function. This is nice but in theory it is still quite similar to the recent contrastive learning approaches.

Typo: eq 9 Z -> H.

**Summary Of The Paper:**

This paper proposes a novel contrastive loss by whitening the embedding vectors in two ways: along the instance dimension and along the feature dimension. The results are comparable to recent works.



**Summary Of The Review:**

The paper is satisfactory.

---

> ### Author Response · Authors · 2021-11-10
> **Response to Reviewer kq6q**
>
> Thanks for your constructive suggestions!
>
> **Main review.** We are sorry to write the objective function in this formulation, which aims to better understanding. However, it seems this formulation does not emphasize the differences between our method and previous methods. Consider the relation of Zero-ICL and SimCLR. Given two set of embeddings (view A and view B) $Z^{A}=\{z_1^{A}, z_2^{A}, \cdots z_n^{A}\}$ and  $Z^{B}=\{z_1^{B}, z_2^{B}, \cdots, z_n^{B}\}$. The later one (SimCLR [1]) calculates the similarity ($z_i^{A}, z_j^B, 1 \leq i \leq N, 1 \leq j \leq N$) of each two embeddings of the two sets, which will takes $\mathcal{O}(N^2)$ time complexity. However, in our Zero-ICL, we first transform the two matrix via whitening to obtain $H^{A}=\{h_1^{A}, h_2^{A}, \cdots, h_n^{A}\}$ and $H^{B}=\{h_1^{B}, h_2^{B}, \cdots, h_n^{B}\}$. Then we only maximize the similarity of $h_i^{A}, h_i^{B}, 1\leq i\leq N$, which only takes $\mathcal{O}(N)$ complexity.
>
> **typos.** Thanks for your good suggestion. we will correct the typo.
>
> [1] Chen T, Kornblith S, Norouzi M, et al. A simple framework for contrastive learning of visual representations[C]//International conference on machine learning. PMLR, 2020: 1597-1607.

---

> > ### Comment · Reviewer_kq6q · 2021-11-14
> > **This linear time argument may not be correct.**
> >
> > since it takes more work to decorrelate $Z^A$ and $Z^B$.

---

> > > ### Author Response · Authors · 2021-11-14
> > > **Response to Reviewer kq6q**
> > >
> > > Thanks for your quick reply!
> > >
> > > Hi, here we only discuss the complexity of the objective function. While from $Z$->$H$, it only takes about one second for $10^{6} \times 10^{5}$ matrix by random SVD ($2048 \times 2048$ in our experiments) [https://research.fb.com/blog/2014/09/fast-randomized-svd/]. Besides, after obtaining the whitening matrix, the $\mathbf{H}$ is calculated by $\mathbf{H} = \mathbf{W}.detach() \mathbf{Z}$, which is a matrix multiplication without gradient backpropagation.
> > >
> > >  We will clarify it in our revised version. Thank you for your reminder!

---

> > > > ### Comment · Reviewer_kq6q · 2021-11-14
> > > > **The complexity.**
> > > >
> > > > Right, but the complexity of matrix multiplication is usually $O(N^3)$.
> > > > But the overall complexity may not be lower than SimCLR.

---

> > > > > ### Author Response · Authors · 2021-11-15
> > > > > **Response to Reviewer kq6q.**
> > > > >
> > > > > Thanks for your quick reply!
> > > > >
> > > > > The complexity of matrix multiplication is usually $O(N^3)$, and by some optimization, it can be reduced to $O(N^{2.xx})$. However, the multiplication of $Z$ and whitening matrix $W$ does not require gradient, which only takes the forward propagation complexity, i.e., $O(N^{2}d)$. Then, consider SimCLR (from line 24 in [https://github.com/leftthomas/SimCLR/blob/master/main.py]), we first concatenate two matrices and use the multiplication, which takes $O(4N^{2}d)$. Even if we don't consider the exp and softmax operations, the backpropagation will also take at least $O(4N^{2}d)$ complexity.
> > > > >
> > > > > In our experiments, Zero-CL with 50 epochs on ImageNet takes about 85154 seconds, Barlow Twins takes about 108561 seconds. SimCLR takes about 112716 seconds. The experiments are conducted on 16 1080Ti GPUs with ResNet-50 backbones.
> > > > >
> > > > > But you are right, if we consider the overall time complexity, the gap between the two methods (SimCLR, Zero-ICL) is only a constant term $O(C N^{2} d)$. However, our method shows a faster convergence rate and is also more robust to hidden dimensions and batch size (see ablation study).

---

### Official Review · Reviewer_AaoY · 2021-11-02

**Correctness:** 3
**Technical Novelty And Significance:** 3
**Empirical Novelty And Significance:** 3
**Recommendation:** 6
**Confidence:** 2

**Main Review:**

Pro:
+ The proposed formulation can be integrated into various methods as a plug-and-play component.
+ The approach is clean and well backed-up both theoretically and experimentally.
+ In-depth ablation study

Cons/Quest:
- How frequent is the mode collapse to trivial solutions in practice for the competing methods? Most of this approaches are quite stable in practice even without using negative samples (i.e. SimSiam).
- What is the actual advantage in wall-clock training time? Does the lower complexity in theory translates well in practice?
- In Table 1, what is the explanation behind the fact that the proposed approach lags behind as the number of iteration increases?
- Can this approach scale well to larger models too?
Minor:
- Section 2, a few paragraphs don't start with a capital letter.

**Summary Of The Paper:**

The paper is concerned with preventing the model collapse in the self-supervised learning scenario using contrastive losses. The main addition is an adaptation to the existing formulations that performs instance or feature whitening, avoiding the use of negative examples than can be expensive to store and to compute the similarity. Experiments on the selected datasets show that the method generally matches the current top-performing methods.

**Summary Of The Review:**

Overall the idea looks clear and efficient, it's not clear however how problematic for the current approaches the collapse is and some additional clarifications could strengthen the work.

---

> ### Author Response · Authors · 2021-11-10
> **Response to Reviewer AaoY**
>
> Thanks for your constructive suggestions!
>
> **Frequency of mode collapse.** We conduct several experiments of evaluating model collapses in Table 4, where our method does not collapse under both w/o and w/ negatives. At least in our numeric experiments (Zero-ICL and Zero-FCL), The collapse frequency is **0**. However, SimCLR [2] and Barlow Twins [1] will completely collapse [3] and dimensionally collapse [3] w/o negatives.
>
> **Wall-clock time.** Since the forward propagation and backpropagation of backbone will take lots of time, here we give spending time of loss calculation to show the difference. For Zero-FCL, it takes 1.61855 seconds over 4k iterations and for Barlow Twins [1], it takes 4.82446 seconds over 4k iterations. Note that the data is from $d=2048$ (one single 1080Ti GPU), and if the hidden dimension is larger, the gap will become larger (due to quadratic and linear order).
>
> **Lags behind as the number of iterations.** Our method converges with a faster rate than previous asymmetric and negative-requiring methods (see Table 3, also discussed in ablation study). Hence, our method gets better results in 100 epochs, while gets lower results in 400 epochs.
>
> **Larger model.** In our main study, we follow Barlow Twins [1], which mainly uses ResNet-50 as backbone. Note that our method reduces the quadratic complexity of objective function to linear, which is more scalable than Barlow Twins [1] and SimCLR [2]. Per your request, we will give results of ResNet152 and ResNet101 as soon as possible (hopefully during the rebuttal period, but subject to the time and resource constraint).
>
> **Typos.** We will carefully check the typos and modify in our next version.
>
> [1] Zbontar J, Jing L, Misra I, et al. Barlow twins: Self-supervised learning via redundancy reduction[J]. arXiv preprint arXiv:2103.03230, 2021.
>
> [2] Chen T, Kornblith S, Norouzi M, et al. A simple framework for contrastive learning of visual representations[C]//International conference on machine learning. PMLR, 2020: 1597-1607.
>
> [3] Jing L, Vincent P, LeCun Y, et al. Understanding Dimensional Collapse in Contrastive Self-supervised Learning[J]. arXiv preprint arXiv:2110.09348, 2021.

---

### Official Review · Reviewer_GfmZ · 2021-11-03

**Correctness:** 3
**Technical Novelty And Significance:** 3
**Empirical Novelty And Significance:** 3
**Recommendation:** 8
**Confidence:** 3

**Main Review:**

This work presents a good extension on the basis of the previous regularization-based SSL method [1], and further, proposes Zero-CL from the instance and feature-wise aspects. The empirical experiment shows only incremental performance gain over the prior arts like Barlow Twins, yet the proposed Zero-CL has an obvious complexity advantage as it's not dependent on the feature dimension.

Some technical doubts regard the Zero-CL:
1. It seems the parameter \lambda regulates the weight of feature/instance terms, is there any intuition behind the setting of lambda=1?

2.  According to what is shown in Figure 3, L_{fea} has a consistent Acc@1 trend with Barlow Twins (larger dimension yields better results) but there does have a seemingly optimal hidden-size (1024 as shown in the Figure). Does this indicate that there's a conflict between the instance/feature-wise objective? In such a case, how could one decide the best-hidden size besides empirical experiences?

[1] Adrien Bardes, Jean Ponce, and Yann LeCun. Vicreg: Variance-invariance-covariance regularization for self-supervised learning. arXiv preprint arXiv:2105.04906, 2021.

**Summary Of The Paper:**

This paper presents a new self-supervised learning objective that aims to further adapt the previous decorrelation-based [1] contrastive learning method [Bardes, 2021], which largely alleviates the trivial solutions in SSL. Compared with prior arts, the proposed Zero-ICL/FCL are constructed from mainly the aspects of instance and feature-wise. In particular, Zero-CL requires no negative samples; feature-wise FCL discards the redundancy term by feature-wise whitening, and the proposed ICL prevents the collapse of contrastive learning effectively. Quantitative evaluations validate that Zero-CL leads to on-par performances with previous state-of-the-art results.



**Summary Of The Review:**

- The paper is very constructed and has a clear motivation, comprehensive ablative studies, and good analytic discussions.

- The only concern from the reviewer is whether the complexity advantage is a sufficient contribution over the previous regularization method [1], given that Zero-CL shows only incremental performance gain.

- The exhibited theoretical analysis of Zero-CL with previous efforts gives direct and clear comparisons.

---

> ### Author Response · Authors · 2021-11-10
> **Response to Reviewer GfmZ**
>
> Thanks for your constructive suggestions!
>
> **Intuition of $\lambda$.** We find that both Zero-ICL and Zero-FCL can work independently, where the performance of the two methods are somewhat similar. Hence, we just simply set the $\lambda=1$. We also try $\lambda=0.5$ and $\lambda=2$. However, we get similar results. Hence, we think Zero-CL is not sensitive to $\lambda$.
>
> **Best hidden size and batch size.** In our experiments, we find Zero-ICL shows similar behavior to SimCLR [3], while Zero-FCL shows similar behavior to Barlow Twins [1]. The two methods are methodologically orthogonal and we try to combine them to seek for the best of the two worlds. Admittedly, in its current form there is no notable improvement (see Table 2 and 3) for which the reason we agree that there still calls for more detailed technical design that may not be able to be covered in this conference paper. Hence, we recommend to use Zero-ICL and Zero-FCL separately.
>
> **Complexity advantage.** VICReg [2] proposes three terms, which are variance, invariance and covariance terms. The invariance term requires $\mathcal{O}(N)$ complexity and the variance term requires $\mathcal{O}(N)$ complexity. However, the covariance term requires $\mathcal{O}(d^2)$ complexity (see the covariance loss in page 16 in <https://arxiv.org/pdf/2105.04906.pdf>). Where Zero-FCL doesn't depend on pair-wise similarity, which only has invariance term. Hence, only $\mathcal{O}(d)$ complexity is required.
>
> [1] Zbontar J, Jing L, Misra I, et al. Barlow twins: Self-supervised learning via redundancy reduction[J]. arXiv preprint arXiv:2103.03230, 2021.
>
> [2] Bardes A, Ponce J, LeCun Y. Vicreg: Variance-invariance-covariance regularization for self-supervised learning[J]. arXiv preprint arXiv:2105.04906, 2021.
>
> [3] Chen T, Kornblith S, Norouzi M, et al. A simple framework for contrastive learning of visual representations[C]//International conference on machine learning. PMLR, 2020: 1597-1607.

---

### Author Response · Authors · 2021-11-16
**General response to reviews**

First of all, we sincerely appreciate all the comments and suggestions of reviewers!

Here we report some modifications in the revised version.

Modify typos $Z->H$ in Eq. 9 (Reviewer kq6q and 1y87).

Modify "complexity" to "complexity of objectives" (Reviewer kq6q).

Add properties and advantages of ZCA literature review (Reviewer 1y87).

Discuss low-rank conditions after Eq. 2 (Reviewer 1y87).

We feel sorry we may not completely solve the concern of Reviewer 1y87 (orthogonal on instance dimension) since it's a standard open problem in mathematics (Equiangular lines). At present, scientists can only give the number of equiangular lines (at most 344 lines) on spaces below the 43 dimensions [1]. However, in contrastive learning, almost all the methods set the output dimension as 256 or larger. However, even in 43 dimensions, if the batch size is lower than 344, the orthogonal solution in one mini-batch can be sub-optimal, since we can ignore the unequal angles due to the uniformity property of InfoNCE [2].

[1] Greaves G R W, Syatriadi J, Yatsyna P. Equiangular lines in Euclidean spaces: dimensions 17 and 18[J]. arXiv preprint arXiv:2104.04330, 2021.

[2] Oord A, Li Y, Vinyals O. Representation learning with contrastive predictive coding[J]. arXiv preprint arXiv:1807.03748, 2018.

---

### Author Response · Authors · 2021-11-26
**Look forward reply to our rebuttal and update**

Dear reviewers, thanks for your comments and constructive suggestions which have inspired us a lot to improve the paper. We are sincerely looking forward to your reply and we could provide more information if needed.

---

### Decision · Program_Chairs · 2022-01-20

**Decision:**

Accept (Poster)

**Comment:**

The initial reviews for this paper were somewhat diverging, however the paper did not receive any significant negative criticism to push it towards below the acceptance threshold. The reviewers have found some minor issues about the paper. Following the reviewer recommendations, the meta reviewer recommends acceptance.